EMBO
Molecular Medicine

# Inhibition of DPP4 activity in humans establishes its *in vivo* role in CXCL10 post-translational modification: prospective placebo-controlled clinical studies

Jérémie Decalf[1,2,†], Kristin V Tarbell[3,†], Armanda Casrouge[1,2], Jeffrey D Price[3], Grace Linder[3], Estelle Mottez[4], Philippe Sultanik[5], Vincent Mallet[5], Stanislas Pol[5], Darragh Duffy[1,2,4,*,‡] & Matthew L Albert[1,2,4,6,**,‡]

## Abstract

Biochemical experiments, animal models, and observational studies in humans all support a role of dipeptidyl peptidase 4 (DPP4) in the N-terminal truncation of CXCL10, which results in the generation of an antagonist form of the chemokine that limits T-cell and NK cell migration. Motivated by the ability to regulate lymphocyte trafficking *in vivo*, we conducted two prospective clinical trials to test the effects of DPP4 inhibition on CXCL10 processing in healthy donors and in chronic hepatitis C patients, a disease in which DPP4 levels are found to be elevated. Participants were treated daily with 100 mg sitagliptin, a clinically approved DPP4 inhibitor. Plasma samples were analyzed using an ultrasensitive single-molecule assay (Simoa) to distinguish the full-length $CXCL10_{1-77}$ from the $NH_2$-truncated $CXCL10_{3-77}$, as compared to the total CXCL10 levels. Sitagliptin treatment resulted in a significant decrease in $CXCL10_{3-77}$ concentration, a reciprocal increase in $CXCL10_{1-77}$, with only minimal effects on total levels of the chemokine. These data provide the first direct evidence that *in vivo* DPP4 inhibition in humans can preserve the bioactive form of CXCL10, offering new therapeutic opportunities for DPP4 inhibitors.

**Keywords** chemokines; clinical study; CXCL10; DPP4; post-translational modifications

**Subject Categories** Microbiology, Virology & Host Pathogen Interaction; Pharmacology & Drug Discovery; Post-translational Modifications, Proteolysis & Proteomics

## Introduction

Chemokines play an essential role in cell migration. Regulation of their activity is particularly important during inflammatory responses, determining the recruitment of immune cells to lymphoid organs or targeting them toward injured tissues (Griffith *et al*, 2014). Post-translational modification of chemokines has been shown to regulate their activity; however, *in vivo* evidence remains limited to observational studies and experimental mouse models (Moelants *et al*, 2013). Dipeptidyl peptidase 4 (DPP4, also known as CD26) is a serine protease capable of removing the first two amino acids of proteins possessing a proline or alanine in the N−terminal penultimate position (Bongers *et al*, 1992). *In vitro* studies have shown that DPP4-mediated N-terminal truncation of the pro-inflammatory chemokine CXCL10 leads to the generation of an antagonist form (Proost *et al*, 2001; Casrouge *et al*, 2011). Moreover, recent *in vivo* work performed in mice has demonstrated that this truncation alters lymphocyte migration and limits infiltration of the tumor parenchyma, a phenomenon that could be reversed using the DPP4 inhibitor sitagliptin (Barreira da Silva *et al*, 2015).

A challenge for studying post-translational modifications of chemokines is the ability to specifically monitor the different protein forms in biological material. To overcome this, we developed

1  The Laboratory of Dendritic Cell Immunobiology, Institut Pasteur, Paris, France
2  INSERM U818, Paris, France
3  Diabetes Endocrinology and Obesity Branch, National Institute of Diabetes and Digestive and Kidney Diseases, National Institutes of Health, Bethesda, MD, USA
4  Center for Human Immunology, Institut Pasteur, Paris, France
5  Département d'Hépatologie, AP-HP, Hôpital Cochin, Université Paris Descartes, INSERM UMS20, Institut Pasteur, Paris, France
6  Department of Cancer Immunotherapy, Genentech, South San Francisco, CA, USA
   *Corresponding author. Tel: +33 1 44 38 93 34; E-mail: darragh.duffy@pasteur.fr
   **Corresponding author. Tel: +33 1 44 38 93 34; E-mail: albertm@pasteur.fr
   †These authors contributed equally to this work
   ‡These authors contributed equally to this work

immunoassays that discriminate the full-length agonist form of CXCL10 (referred to as $CXCL10_{1-77}$, or long CXCL10) from the $NH_2$-truncated form generated by DPP4 cleavage (referred to as $CXCL10_{3-77}$, or short CXCL10) (Casrouge *et al*, 2012). Accordingly, we were able to show that elevated levels of short CXCL10 were associated with an increased DPP4 activity, both being negative predictors for viral clearance in chronic and acute hepatitis C (HCV) patients (Casrouge *et al*, 2011; Ragab *et al*, 2013; Riva *et al*, 2014). Taking advantage of the Simoa technology (Rissin *et al*, 2010), we have now developed ultrasensitive immunoassays with these antibodies, making possible the quantification of functional forms of CXCL10 in the plasma from healthy individuals.

To provide a proof of concept for DPP4 inhibition as a means to protect full-length CXCL10, we conducted a randomized placebo-controlled study in healthy individuals, with the primary scientific objective being the assessment of DPP4 inhibition on immunological parameters. A prior report on this cohort demonstrated an effective DPP4 inhibition and an increased concentration of active glucagon-like peptide 1, one of the key substrates implicated in insulin resistance (Price *et al*, 2013). In parallel, we undertook an investigational study to explore how sitagliptin treatment affects virus-induced CXCL10 in chronic HCV patients (cHCV), a clinical setting in which levels of both DPP4 and $CXCL10_{3-77}$ have been shown to be elevated (Ragab *et al*, 2013). The findings from this report establish the basis for repositioning DPP4 inhibitors as a potential immunotherapy.

## Results

To quantify CXCL10 agonist and antagonist forms in healthy individuals, we implemented our unique immunoassays on the Simoa platform (Fig 1A). Using these assays, we determined median levels of total CXCL10 to be 60 pg/ml in healthy individuals (Fig 1B and C, SV and D0 time points), within the range of what has been previously reported using other techniques (Butera *et al*, 2005; Duffy *et al*, 2014). Long CXCL10 remained undetectable in most individuals, whereas short CXCL10 was detected in 30 out of 36 subjects (83%) with concentrations ranging from 2 to 75 pg/ml, suggesting active *in vivo* chemokine processing in healthy individuals. Importantly, all the plasma samples used for the analysis of CXCL10 levels were collected in tubes containing a DPP4 inhibitor to avoid potential extracorporeal CXCL10 processing.

In order to explore the role of DPP4 in CXCL10 truncation *in vivo*, we monitored the levels of short, long, and total CXCL10 in healthy individuals receiving a 28-day course of placebo or sitagliptin (Fig 1B and C). DPP4 inhibition in individuals receiving sitagliptin was confirmed by monitoring plasma DPP4 activity, which was previously published (Price *et al*, 2013) and showed an inhibition ranging from 9 to 80% (Fig EV1). We observed that 3 days after the onset of sitagliptin treatment, the concentrations of short CXCL10 dropped significantly in individuals receiving sitagliptin, but remained stable in donors receiving the placebo. Effect size analysis supported the strong impact of sitagliptin on short CXCL10 when compared to pre-therapy levels ($d = 0.92$). As detailed in Fig EV1, some donors showed sporadic increases in short CXCL10 during sitagliptin therapy, possibly reflecting a partial recovery of DPP4 activity (Herman *et al*, 2005). Moreover, three of the 27 donors showed elevated levels of short CXCL10 during sitagliptin treatment,

reflecting natural human variability in response to the treatment. Of note, despite the presence of short CXCL10, two of these three donors showed a strong DPP4 inhibition.

Interestingly, the inhibition of DPP4 was associated with the preservation of long CXCL10, as indicated by an increase in concentration compared to pre-therapy ($d = 0.27$), although these data have to be interpreted cautiously due to a large number of patients in which long CXCL10 remained undetectable. That said, this increase was observed at week 2 of treatment, but not at day 3 post-treatment initiation, perhaps reflecting a low level of newly secreted CXCL10 in healthy individuals. Of note, the levels of total CXCL10 were found to be slightly decreased during sitagliptin treatment compared to pre-therapy levels ($d = 0.33$), suggesting a limited biological impact of sitagliptin on total CXCL10 levels. That said, the strong impact of sitagliptin on short CXCL10 indicated that we selectively altered processing, rather than the production of the chemokine. Finally, the modulation of short and long CXCL10 was stable during the 28-day course of sitagliptin, returning to pre-therapy concentrations once treatment was terminated. These data provide direct evidence that DPP4 inhibition impacts *in vivo* N-terminal truncation of CXCL10.

The data obtained in healthy individuals suggest that CXCL10 processing by DPP4 is a rapid event, as it was strongly affected 72 h after the onset of sitagliptin therapy. Therefore, we assessed how sitagliptin might affect higher levels of CXCL10, a hallmark of inflammatory diseases (Van Raemdonck *et al*, 2015). To do so, we monitored CXCL10 forms in three cHCV patients receiving sitagliptin treatment (Fig 1D). Of note, as detailed in the study design section, additional patients could not be recruited for ethical reasons. As previously described (Casrouge *et al*, 2011), cHCV patients showed elevated concentrations of short, long, and total CXCL10 compared to healthy individuals (D0 time points). Interestingly, sitagliptin treatment led to a decrease in short CXCL10 and an increase in long CXCL10 (Fig 1D), a trend similar to what we observed in healthy individuals. Although striking, the impact of sitagliptin on CXCL10 forms did not influence HCV viral loads over the period monitored (Fig EV2).

## Discussion

Overall, the finding that the agonist form $CXCL10_{1-77}$ was undetectable in the majority of healthy individuals indicates that CXCL10 and likely other chemokines are rapidly catabolized by DPP4. The corollary to this observation is that long CXCL10 may be considered as a marker of recently produced CXCL10. Moreover, the lower levels of long and short forms compared to the total plasma concentration of CXCL10 indicate that additional processing of the protein is probably occurring *in vivo*. Other proteases, such as matrix metalloproteinases (Van den Steen *et al*, 2003), have been shown to target CXCL10, but their activity *in vivo* and the impact on chemokine function remain unknown (see Mortier *et al* (2008) for review of subject). Action of other N-terminal aminoproteases could also explain the trimming of $CXCL10_{3-77}$, acting after DPP4 removes the penultimate proline residue, as shown in *in vitro* biochemical studies using CXCL11 (Proost *et al*, 2007).

Our findings expose a broader putative *in vivo* role of DPP4 in the regulation of cell trafficking. Notably, other chemokine substrates of

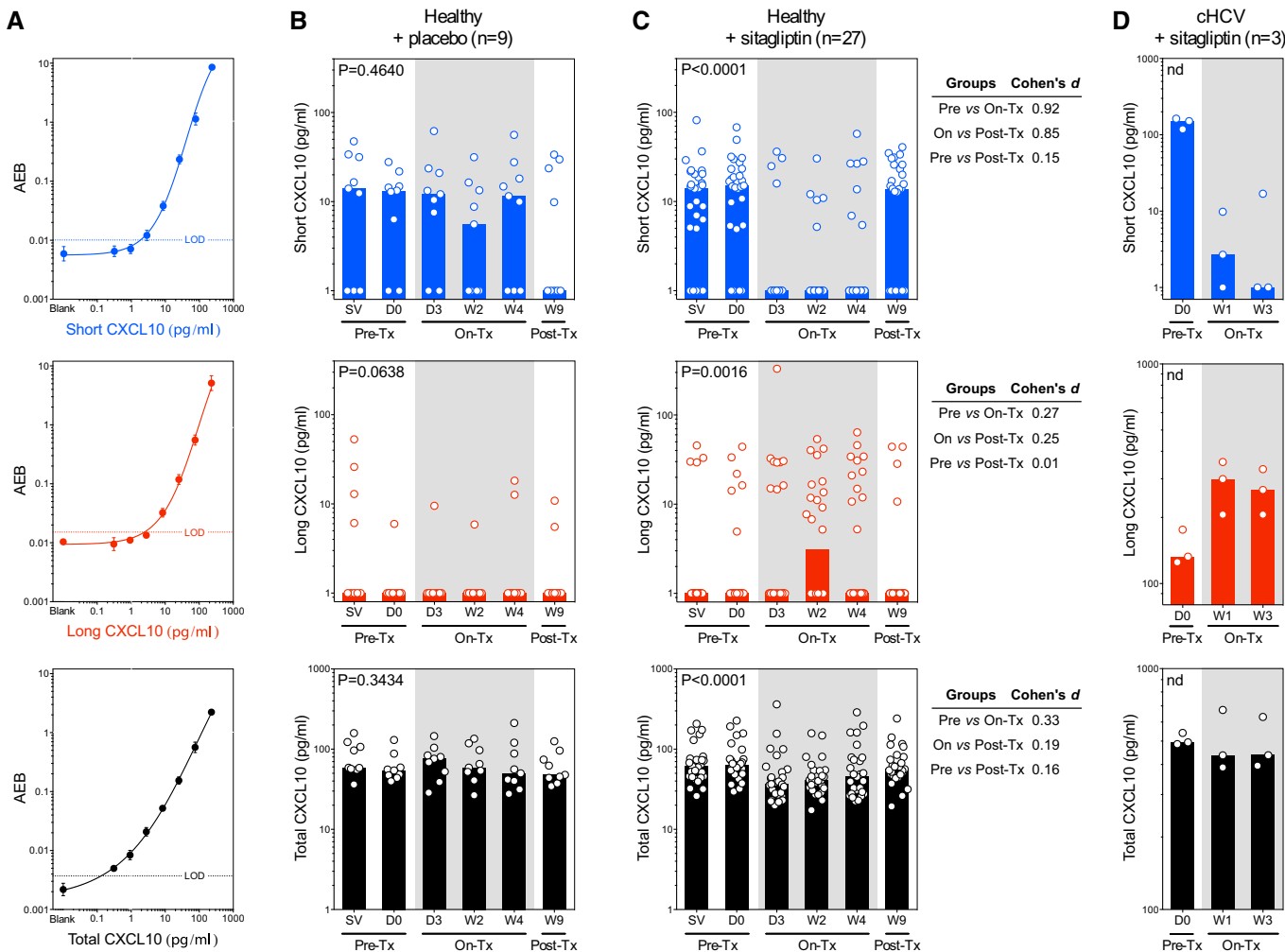

**Figure 1.  Sitagliptin treatment limits DPP4-mediated N-terminal truncation of CXCL10.**

A    CXCL10 assay characteristics. Standard curves of the short, long, and total CXCL10 Simoa assays are shown. For each curve, the limit of detection (LOD) defined as blank+3SD is shown as horizontal lines. The LOD was 1.7 pg/ml for long and short CXCL10 assays and 0.22 pg/ml for total CXCL10. Standard curves were fitted using the 4-parameter logistic nonlinear regression model. Samples reporting a signal below the LOD were replaced with 1 pg/ml for short and long CXCL10.

B–D  Plasma from healthy individuals receiving (B) placebo ($n = 9$) or (C) sitagliptin ($n = 27$) was analyzed by Simoa. Blood samples were collected at screening visit (SV) and day 0 (D0) before treatment; at day 3 (D3), day 14 (W2), and day 28 (W4) under treatment; and 5 weeks after treatment interruption (W9). (D) Plasma from chronic HCV patients receiving sitagliptin ($n = 3$) was collected before (D0) and weekly during sitagliptin treatment (W1 & W3). Antagonist CXCL10$_{3-77}$ (short CXCL10, in blue), agonist CXCL10$_{1-77}$ (long CXCL10, in red), and total CXCL10 (in black) levels are shown. Each dot represents a donor, and bars are at the median. Gray areas highlight the period under placebo or sitagliptin treatment. Statistical analysis of (B) and (C) was performed using nonparametric Friedman's test, ns: nonsignificant, **$P < 0.01$, ****$P < 0.001$. For (C), additional size effect analysis was performed and Cohen's $d$ values are reported. No statistical analysis was performed in (D) due to sample size, nd: nondetermined.

DPP4 have been defined as key mediators of stem cell migration. In mouse studies, DPP4 inhibition has been shown to improve the engraftment of CD34$^+$ stem cells following umbilical cord blood transplantation, with the mechanism of action being the protection of agonist forms of CXCL12 (Farag *et al*, 2013). While our study does not directly test a role for altered post-translational modification of CXCL12, the similar biological processes would suggest a direct impact on CXCR4-mediated leukocyte migration.

From a therapeutic perspective, our results establish a path toward the secondary use of DPP4 inhibitors. Indeed, our recent

**Table 1.  Clinical information.**

| Category | Treatment | Number of subjects | Gender (F/M) | Age - median (range) | HCV genotype |
|---|---|---|---|---|---|
| Healthy | Placebo | 9 | 6/3 | 27 (20–61) | – |
| Healthy | Sitagliptin | 27 | 16/11 | 36 (19–53) | – |
| cHCV | Sitagliptin | 3 | 2/1 | 41 (38–55) | 4 |

mouse studies have shown that preserving agonist $CXCL10_{1-77}$ increases lymphocyte infiltration into tumor parenchyma and results in an enhanced tumor immunity (Barreira da Silva *et al*, 2015). Thus, these drugs may offer an opportunity to increase the lymphocyte migration in settings such as chronic infection and cancer. As DPP4 inhibitors are a class of widely used therapeutics for the management of type II diabetes, they are considered to be safe and well tolerated.

In sum, our work provides the first *in vivo* experimental evidence of CXCL10 processing in humans and supports the clinical testing of sitagliptin as either an antitumor or autoimmune therapy. Further work will be required to better understand the impact of DPP4 inhibition *in vivo* and to evaluate how other forms of chemokine post-translational modifications could influence inflammatory responses.

## Materials and Methods

### Study design and protocols

The cohort of healthy individuals receiving sitagliptin or placebo (NCT00813228) has been previously described. This study was a double-blind, randomized trial approved by the institutional review board of NIDDK (Price *et al*, 2013). Participants received 100 mg of sitagliptin ($n = 27$) or placebo ($n = 9$) once daily for 28 days, and the blood samples were collected longitudinally. Chronic HCV patients receiving sitagliptin ($n = 3$) were recruited as part of the INSERM-sponsored C10-54 trial (NCT01567540). The three patients were chronically infected with HCV genotype 4 and were receiving pegylated-interferon alpha + ribavirin therapy when they enrolled in the study. These patients were considered difficult to treat, as they had previously failed to achieve early virological response (EVR) as defined by a two-log reduction in virus at 12 weeks post-treatment. Sitagliptin treatment (100 mg daily) was given for 3 weeks and the blood samples were collected longitudinally. Of note, the C10-54 trial was terminated prematurely due to the approval of sofosbuvir in France for HCV genotype 4-infected patients. The ethical implications that arose from the availability of this novel highly efficacious antiviral therapy prevented the continued recruitment of additional cHCV patients in our experimental clinical study. The respective studies were approved by the institutional review boards of the NIH and INSERM. All participants gave written informed consent prior to inclusion in the study, conformed to the principles set out in the WMA Declaration of Helsinki and the Department of Health and Human Services Belmont Report. Clinical characteristics are summarized in Table 1. Plasma was collected in BD P700 tubes, containing ethylenediamine tetraacetic acid (EDTA) and a DPP4 inhibitor to prevent extracorporeal cleavage of CXCL10. Plasma collected in sodium heparin tubes was used for monitoring DPP4 activity. Samples were stored at −80°C until analysis.

### CXCL10 quantification

Plasma concentration of total (R&D clone 33036), long ($CXCL10_{1-77}$), and short ($CXCL10_{3-77}$) CXCL10 was measured using Simoa technology (Quanterix). Simoa assays were carried out as

### The paper explained

**Problem**

The N-terminal truncation of CXCL10 by DPP4 results in the generation of an antagonist form of the chemokine that limits T-cell and NK cell migration toward infectious or tumor sites. In this work, we studied whether DPP4 inhibition *in vivo* by sitagliptin could limit the generation of the antagonist form of CXCL10 in humans, which could have the potential to boost T-cell and NK cell migration in certain pathological contexts.

**Results**

Participants were treated daily with 100 mg sitagliptin, and plasma samples were analyzed using an ultrasensitive single-molecule assay (Simoa) to distinguish levels of the agonist ($CXCL10_{1-77}$), antagonist ($CXCL10_{3-77}$), and total CXCL10 forms. Sitagliptin treatment resulted in a significant decrease in antagonist CXCL10 concentration and a reciprocal increase in the agonist form $CXCL10_{1-77}$ compared to placebo controls.

**Impact**

Our data provide the first *in vivo* evidence that DPP4 inhibition in humans can preserve the bioactive form of CXCL10. This offers new therapeutic opportunities for DPP4 inhibitors, which could be relevant for the development of novel cancer immunotherapies aiming at restoring immune cell migration.

previously described (Meissner *et al*, 2015), and additional optimization was conducted for this study; standard curves are presented in Fig 1A.

### Statistical analysis

Differences in CXCL10 forms in healthy donors receiving either sitagliptin or placebo were assessed by nonparametric Friedman's test. No statistical analysis was performed on HCV patients due to the available sample size ($n = 3$). For healthy donors receiving sitagliptin, size effect analysis using Cohen's *d* was performed to assess the biological significance of modified CXCL10 levels. Three groups of samples were compared; the values were obtained pre-therapy (SV, D0), during therapy (D3, W2, and W4), and post-therapy (W9). Cohen's *d* estimates the biological significance of statistically significant differences to be small ($d \approx 0.2$), medium ($d \approx 0.5$), or large ($d \approx 1$).

**Expanded View** for this article is available online.

### Acknowledgements

This work was supported by the Agence nationale de recherches sur le sida et les hépatites virales (ANRS) and the Ligue nationale contre le cancer and by the Intramural Research Program of the National Institute of Diabetes and Digestive and Kidney Diseases (NIDDK).

### Author contributions

JD and AC performed the experiments; JD, KVT, AC, MLA, and DD analyzed the data; JP, GL, EM, PS, VM, SP, KVT, MLA, and DD designed and executed the clinical studies; JD, DD, and MLA wrote the manuscript.

### Conflict of interest

The authors declare that they have no conflict of interest.

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
