## [Review Process File · EMBO Molecular Medicine]

Inhibition of DPP4 activity in humans establishes its in vivo role in CXCL10 post-translational modification: prospective placebo-controlled clinical studies

J r mie Decalf , Kristin V Tarbell , Armanda Casrouge, Jeffrey D Price, Grace Linder, Estelle Mottez, Philippe Sultanik, Vincent Mallet, Stanislas Pol, Darragh Duffy and Matthew L Albert

Corresponding author: Darragh Duffy and Matthew L Albert, Institut Pasteur

Review timeline:

Submission date:	14 December 2015
Editorial Decision:	14 January 2016
Appeal:	29 January 2016
Editorial Decision:	04 February 2016
Revision received:	19 February 2016
Accepted:	16 March 2016

Transaction Report:

Editor: C line Carret

1st Editorial Decision

14 January 2016

Thank you for the submission of your research manuscript to our editorial office. We have now received the enclosed reports on it.

As you will see, both referees find the study to be clinically important, but unfortunately too preliminary. Of critical relevance for our scope, both referees regret the limited clinical relevance as not enough HepC patients were enrolled in the trial to make the data significant enough to be published in EMBO Molecular Medicine.

Given the nature of these criticisms, the amount of work likely to be required to address them, and the fact that, EMBO Molecular Medicine can only invite revision of papers that receive enthusiastic support from a majority of referees especially in light of our single round of revision rule, I am afraid that we do not feel it would be productive to call for a revised version of your manuscript at this stage and therefore we cannot offer to publish it.

However, should you be able, at a later stage, to increase the number of patients and provide a better description and analysis of the results as suggested by the reviewers, we would not be opposed to consider a new manuscript on the same topic. To be completely clear, however, I would like to stress that if you were to send a new manuscript this would be treated as a new submission rather than a revision and would be reviewed afresh, in particular with respect to the literature and the novelty of your findings at the time of resubmission. If you decide to follow this route, please make sure to nevertheless upload a letter of response to the referees' comments.

At this stage though, I am sorry to have to disappoint you. I nevertheless hope, that the referee comments will be helpful in your continued work in this area and I thank you for considering EMBO Molecular Medicine.

***** Reviewer's comments *****

Referee #1 (Remarks):

Dipeptidyl peptidase 4 (DPP4) catalyzes the cleavage of two N-terminal amino acids from glucagon-like peptide 1 and several other proteins, including the chemokine CXCL10. Sitagliptin is a DPP4 inhibitor approved for treatment of type II diabetes. Many groups have reported that CXCL10 is increased in the blood of patients with chronic hepatitis C virus infection (cHCV), and the Albert group reported in 2011 that elevated CXCL10 was associated with the presence of a processed form similar to what would be expected after DPP4 cleavage. This shortened form of CXCL10 is believed to antagonize the chemotactic activity of the full-length form of CXCL10.

In this brief report, Decalf et al present the effects of sitagliptin treatment on CXCL10 levels and processing in 27 healthy subjects and 3 cHCV patients treated with sitagliptin, comparing them to 9 healthy subjects treated with a placebo. As predicted, sitagliptin treatment reduced DPP4-mediated processing of CXCL10 in both healthy and cHCV subjects. This result supports the claim that DPP4 activity contributes to CXCL10 cleavage in vivo. These preliminary human subject results are consistent with those recently published by the senior author in a mouse model (Nature Immunology 2015). The importance of the human result is that it provides a basis for human clinical trials in settings in which excess DPP4 activity and chemokine processing may impair immune function.

Specific comments

1. Please indicate whether the cHCV patients included in this study were on other antiviral treatment.
2. I recognize that this report is quite preliminary. However, the inclusion of cHCV patients in this study begs the question of whether and how sitagliptin affects HCV viral load. This information would add to the novelty and interest of the report.
3. In the abstract, lines 33-35, the authors write, "Sitagliptin treatment resulted in a significant decrease of CXCL103-77 concentration and a reciprocal increase in CXCL101-77, without affecting the total levels of the chemokine." This statement does not reflect the data in Figure 1C, showing a statistically significant reduction in total CXCL10 in healthy subjects during sitagliptin treatment. Indeed, in line 96 of the paper, the authors write, "Of note, levels of total CXCL10 were slightly decreased during sitagliptin treatment..."
4. Figure 1C shows that sitagliptin has little or no effect on short CXCL10 levels in a number of healthy subjects. The authors may wish to comment on this. Did the same subjects show higher short CXCL10 levels at more than one time point?
5. Were power calculations performed to support the usefulness of a study with only 3 cHCV patients?

Referee #2 (Comments on Novelty/Model System):

Technical criticisms are below.

It is novel, but appears to be incomplete.

Medical impact is impaired by little disease data.

Referee #2 (Remarks):

CXCL10 (IP10) is an important substrate of the DPP4 enzyme and an important chemoattractant of T and NK cells. Activated T and NK cells have high levels of cell surface DPP4 (CD26). Key authors of this ms derive from a research group that has shown in previous papers that DPP4 is an important enzyme in the degradation of CXCL10 in mice, and that DPP4 converts human CXCL10 into an antagonist of its cognate receptor CXCR3 in vitro. The enzyme activities of human

and mouse DPP4 are believed to be the same. The biology of CXCL10 might differ between species.

The concentrations of DPP4 are elevated in human plasma in many chronic diseases, including HCV patients and some categories of cancer patients. Several human cancers have been associated with decreased rather than increased circulating DPP4 levels [review Gorrell 2001 Scand J Immunol].

This research group has shown in previous papers that the levels in human plasma of DPP4 and the DPP4-cleaved form of CXCL10 tend to correlate.

Comments:

1. The primary question that needs a rigorous investigation in such studies as this ms is where the processing of CXCL10 occurs. It could occur close to CXCR3-positive cells, in tissues, in circulating blood, or in blood or plasma after the blood is collected.

The most important, and readily accessible, ancillary question is whether the inhibition of CXCL10 degradation occurred entirely *in vivo* or to what extent the medicinally-derived DPP4 inhibitor that would have been in the blood at the time of blood collection caused the observed results.

To address this issue, Sitagliptin should have been added to all blood at the moment of blood collection in order to prevent any *ex-vivo* CXCL10 digestion. This is a confounder of data interpretation. DPP4 is very active at ambient temperature and has activity at 4 degrees, but not after freezing, so there is probably a correlation between the length of time after blood collection until the blood or plasma is frozen and the concentration of short-CXCL10.

An experiment should be performed to determine effects on CXCL10 of collecting blood into Sitagliptin. This data should be added to this ms.

2. How many patients actually consumed the prescribed sitagliptin drug? Please present data on the concentrations of sitagliptin in the plasma samples. Such data will inform the reader on whether the concentrations of short/long CXCL10 that were measured correlate with the concentrations of sitagliptin in the blood samples.

3. Early work on DPP4 measurements in human blood found that there are various causes of variability, including age, lower in women, increase in summer season, and decrease with depression which might or might not have been related to anti-depressant drugs. Were these parameters controlled in this study? Many HCV patients are depressed and prescribed anti-depressant drugs.

References:

a. Maes, M., ScharpÈ, S., De Meester, I., Goossens, P., Wauters, A., Neels, H., Verkerk, R., De Meyer, F., P, D.H., Peeters, D., et al. 1994. Components of biological variation in prolyl endopeptidase and dipeptidyl-peptidase IV activity in plasma of healthy subjects. *Clin. Chem.* 40:1686-1691.

b. Maes, M., De Meester, I., Verkerk, R., Demedts, P., Wauters, A., Vanhoof, G., Vandoolaege, E., Neels, H., and ScharpÈ, S. 1997. Lower serum dipeptidyl peptidase IV activity in treatment resistant major depression - relationships with immune-inflammatory markers. *Psychoneuroendocrinology* 22:65-78.

c. Maes, M., Goossens, F., Lin, A., De Meester, I., Van Gastel, A., and ScharpÈ, S. 1998. Effects of psychological stress on serum prolyl endopeptidase and dipeptidyl peptidase IV activity in humans: higher serum prolyl endopeptidase activity is related to stress-induced anxiety. *Psychoneuroendocrinology* 23:485-495.

d. Durinx, C., Neels, H., Van der Auwera, J.C., Naelaerts, K., ScharpÈ, S., and De Meester, I. 2001. Reference values for plasma dipeptidyl-peptidase IV activity and their association with other laboratory parameters. *Clin. Chem. Lab. Med.* 39:155-159.

4. Are the statements in the discussion justified regarding the significance of this study? There is probably some over-statement. This research group has shown that in mice DPP4 inhibition can increase the influx of lymphocytes into tumours. Is DPP4 inhibition likely to also increase systemic inflammation?, and thus exacerbate patients with inflammatory disorders. Several human cancers and rheumatoid arthritis have been associated with decreased rather than increased circulating DPP4 levels. The line of argument regarding the potential role of DPP4 inhibitors in autoimmune disease is unclear since long CXCL10, which is preserved by DPP4 inhibitors, is immunostimulatory.

5. Overall, this study is incomplete. For instance, the detection range which is defined as between blank+3SD and blank+10SD only encompasses a single data point on the standard curves which

goes against the principle of interpolation of sample data from standards data.

Since the current study advocates for the potential application of sitagliptin in a therapeutic context, a rigorous examination of drug action in disease is warranted and unfortunately the disease group comprised only 3 patients which understandably did not permit any determination of statistical significance.

The authors propose that long CXCL10 increases at week 2 but the purported increase is detected in week 2 only and is not maintained into week 4 of the treatment window. Similarly, it is disappointing that the disease group was not monitored past week 2 and there is no data to support the reversibility of drug action upon withdrawal of therapy in this group.

Although the authors highlight that the levels of CXCL10 decrease during sitagliptin treatment, there also appears to be increased spread in the data during the treatment period [patient non-compliance?]. Inclusion of SD error bars are required for clarification.

6. Statistics was appropriate. The use of Friedmans test was likely necessary because the data was non-parametric.

Appeal

29 January 2016

Thank you for your time on the phone this week to discuss our recent submission. We are following-up to reiterate the clinical importance of our study, a point that was highlighted by both referees, and to kindly ask that you reconsider your position and permit us to submit a revised version of the manuscript.

As discussed, we are sensitive to the request to increase this size in the Hepatitis C study, however this is no longer possible from an ethical standpoint. Despite the low number of patients enrolled, we feel an obligation to share the work with the community. This desire is based on the generous participation by the patients who volunteered to be part of the study and by the considerable effort and investment in this investigator lead interventional clinical study. To provide some context, we conceived this trial during the roll out of new direct acting anti-virals. While we could justify the opening of the trial in 2012 and the enrolment of HCV genotype 4 patients as they were not benefiting from the first generation NS3 inhibitors, the second generation of DAAs (Sofosbuvir) were approved earlier than anticipated and we felt as if we could not withhold these highly efficacious new treatments. As a result we terminated our Phase 1b trial early despite having enrolled only 3 patients. While a minor set back for our DPP4 program, the data is conclusive – 3 of 3 patients showed a diminished concentration of the short form of CXCL10, which clearly supports the healthy subject study data presented herein.

We believe our study provides new important information to the community and as such merits publication in a quality journal such as EMBO Molecular Medicine. In addition to this we wish to highlight that we are able to fully address all other points raised by the reviewers as detailed below (please see a draft version of our point-by-point reply indicating what we can do to address the reviewers' concerns). We believe that the publication of these results will generate interest and support for additional future studies that will build upon the strategy of inhibiting DPP4 activity potentially leading to new therapeutic approaches.

Referee #1 (Remarks):

Dipeptidyl peptidase 4 (DPP4) catalyzes the cleavage of two N-terminal amino acids from

glucagon-like peptide 1 and several other proteins, including the chemokine CXCL10. Sitagliptin is a DPP4 inhibitor approved for treatment of type II diabetes. Many groups have reported that CXCL10 is increased in the blood of patients with chronic hepatitis C virus infection (cHCV), and the Albert group reported in 2011 that elevated CXCL10 was associated with the presence of a processed form similar to what would be expected after DPP4 cleavage. This shortened form of CXCL10 is believed to antagonize the chemotactic activity of the full-length form of CXCL10.

In this brief report, Decalf *et al* present the effects of sitagliptin treatment on CXCL10 levels and processing in 27 healthy subjects and 3 cHCV patients treated with sitagliptin, comparing them to 9 healthy subjects treated with a placebo. As predicted, sitagliptin treatment reduced DPP4-mediated processing of CXCL10 in both healthy and cHCV subjects. This result supports the claim that DPP4 activity contributes to CXCL10 cleavage *in vivo*. These preliminary human subject results are consistent with those recently published by the senior author in a mouse model (*Nature Immunology* 2015). The importance of the human result is that it provides a basis for human clinical trials in settings in which excess DPP4 activity and chemokine processing may impair immune function.

- We thank the reviewer for highlighting the importance of our results and their potential as a basis for future clinical trials.

Specific comments

1. Please indicate whether the cHCV patients included in this study were on other antiviral treatment.

- The patients included in our study were receiving pegylated Inteferon + ribavirin, the standard of care for cHCV g4 at the time of recruitment, in addition to the experimental dose of sitagliptin. If given the opportunity to revise the manuscript, these details will be added to the methods and materials. As detailed in the cover letter, the approval of second-generation direct acting antivirals occurred much earlier than anticipated, with efficacy in chronic HCV g4 infected patients, thus obliging us to terminate our clinical trial.

2. I recognize that this report is quite preliminary. However, the inclusion of cHCV patients in this study begs the question of whether and how sitagliptin affects HCV viral load. This information would add to the novelty and interest of the report.

- The patients that were enrolled had failed prior type I IFN therapy and thus are considered hard-to-treat patients. In all three instances, patients failed to achieve an early virological response (EVR) as defined by a two-log reduction in virus at 12 weeks post-treatment. We agree with the reviewer that this information is of interest and we can provide this data in a revised version of the study.

3. In the abstract, lines 33-35, the authors write, "Sitagliptin treatment resulted in a significant decrease of CXCL103-77 concentration and a reciprocal increase in CXCL101-77, without affecting the total levels of the chemokine." This statement does not reflect the data in Figure 1C, showing a statistically significant reduction in total CXCL10 in healthy subjects during sitagliptin treatment. Indeed, in line 96 of the paper, the authors write, "Of note, levels of total CXCL10 were slightly decreased during sitagliptin treatment..."

- We thank the reviewer for highlighting this discrepancy for which we apologize and we can modify the manuscript accordingly. In order to assess the magnitude of the biological response to sitagliptin, we propose to include an effect-size analysis as detailed below:

Short CXCL10		Long CXCL10		Total CXCL10	
Groups	Cohen's d	Groups	Cohen's d	Groups	Cohen's d
Pre-Tx vs On-Tx	0.92	Pre-Tx vs On-Tx	0.27	Pre-Tx vs On-Tx	0.33
On-Tx vs Post-Tx	0.85	On-Tx vs Post-Tx	0.25	On-Tx vs Post-Tx	0.19
Pre-Tx vs Post-Tx	0.15	Pre-Tx vs Post-Tx	0.01	Pre-Tx vs Post-Tx	0.16

To perform this analysis, we compared the CXCL10 concentrations measured in all Pre-, On- and Post-sitagliptin samples. As shown, short CXCL10 was strongly affected by sitagliptin treatment and returned to baseline levels once therapy stopped. This phenotype was also true for long CXCL10, although to a lower extent. Size effect analysis showed a moderate influence of sitagliptin on total CXCL10 levels, supporting our interpretation that although long and short CXCL10 are affected by therapy, total levels of CXCL10 are mildly changed. We have included this information to the manuscript.

4. Figure 1C shows that sitagliptin has little or no effect on short CXCL10 levels in a number of healthy subjects. The authors may wish to comment on this. Did the same subjects show higher short CXCL10 levels at more than one time point?

- We thank the reviewer for highlighting this point. The 3 donors in question showed consistently high levels of short CXCL10 at the 3 time points examined (shown in graphic below). Of these 3 individuals only 1 showed a complete lack of DPPIV inhibition. This information can be added to the revised manuscript.

5. Were power calculations performed to support the usefulness of a study with only 3 cHCV patients?

- The original study was approved for an initial recruitment of 10 cHCV patients as the primary objective was safety assessment of the combined therapies. As detailed above the study was prematurely stopped due to ethical reasons related to approval of highly efficacious anti-viral therapies.

Referee #2 (Comments on Novelty/Model System):

Technical criticisms are below.

It is novel, but appears to be incomplete.

Medical impact is impaired by little disease data.

- We thank the reviewer for highlighting the novelty of our findings.

Referee #2 (Remarks):

CXCL10 (IP10) is an important substrate of the DPP4 enzyme and an important chemoattractant of T and NK cells. Activated T and NK cells have high levels of cell surface DPP4 (CD26).

Key authors of this ms derive from a research group that has shown in previous papers that DPP4 is an important enzyme in the degradation of CXCL10 in mice, and that DPP4 converts human CXCL10 into an antagonist of its cognate receptor CXCR3 in vitro. The enzyme activities of human and mouse DPP4 are believed to be the same. The biology of CXCL10 might differ between species.

The concentrations of DPP4 are elevated in human plasma in many chronic diseases, including HCV patients and some categories of cancer patients. Several human cancers have been associated with decreased rather than increased circulating DPP4 levels [review Gorrell 2001 Scand J Immunol]. This research group has shown in previous papers that the levels in human plasma of DPP4 and the DPP4-cleaved form of CXCL10 tend to correlate.

Comments:

1. The primary question that needs a rigorous investigation in such studies as this ms is where the processing of CXCL10 occurs . It could occur close to CXCR3-positive cells, in tissues, in circulating blood, or in blood or plasma after the blood is collected.

The most important, and readily accessible, ancillary question is whether the inhibition of CXCL10 degradation occurred entirely in vivo or to what extent the medicinally-derived DPP4 inhibitor that would have been in the blood at the time of blood collection caused the observed results.

To address this issue, Sitagliptin should have been added to all blood at the moment of blood collection in order to prevent any ex-vivo CXCL10 digestion. This is a confounder of data interpretation. DPP4 is very active at ambient temperature and has activity at 4 degrees, but not after freezing, so there is probably a correlation between the length of time after blood collection until the blood or plasma is frozen and the concentration of short-CXCL10.

An experiment should be performed to determine effects on CXCL10 of collecting blood into Sitagliptin. This data should be added to this ms.

- The reviewer is correct to highlight this important point. All blood samples in our studies were collected in P700 tubes, which contain DPP4 inhibitors. This information is described in the materials and methods. Therefore our measurements reflect *in vivo* CXCL10 forms and not *ex vivo* (extracorporeal) cleavage events.

2. How many patients actually consumed the prescribed sitagliptin drug? Please present data on the concentrations of sitagliptin in the plasma samples. Such data will inform the reader on whether the concentrations of short/long CXCL10 that were measured correlate with the concentrations of sitagliptin in the blood samples.

- Data on DPP4 inhibition was previously presented (Price *et al*, CEI 2013) for the healthy donors, which is why it was not included in this paper. Of the 3 donors that consistently had higher levels of short CXCL10, one did not show DPP4 inhibition, perhaps reflecting human variability with respect to response to treatment. We can provide additional information in the revised manuscript if given the chance to revise the study. For cHCV patients, we note the higher concentrations of DPP4 in the plasma. This, combined with the

rapid *in vivo* clearance of sitagliptin made it difficult to accurately determine the impact on enzyme activity (Herman GA et al, *Clin. Pharmacol. Ther.* 78, 675–688, 2005).

3. Early work on DPP4 measurements in human blood found that there are various causes of variability, including age, lower in women, increase in summer season, and decrease with depression which might or might not have been related to anti-depressant drugs. Were these parameters controlled in this study? Many HCV patients are depressed and prescribed anti-depressant drugs.

References:

a. Maes, M., Scharpé, S., De Meester, I., Goossens, P., Wauters, A., Neels, H., Verkerk, R., De Meyer, F., P, D.H., Peeters, D., et al. 1994. Components of biological variation in prolyl endopeptidase and dipeptidyl-peptidase IV activity in plasma of healthy subjects. *Clin. Chem.* 40:1686-1691.

b. Maes, M., De Meester, I., Verkerk, R., Demedts, P., Wauters, A., Vanhoof, G., Vandoolaeghe, E., Neels, H., and Scharpé, S. 1997. Lower serum dipeptidyl peptidase IV activity in treatment resistant major depression - relationships with immune-inflammatory markers. *Psychoneuroendocrinology* 22:65-78.

c. Maes, M., Goossens, F., Lin, A., De Meester, I., Van Gastel, A., and Scharpé, S. 1998. Effects of psychological stress on serum prolyl endopeptidase and dipeptidyl peptidase IV activity in humans: higher serum prolyl endopeptidase activity is related to stress-induced anxiety. *Psychoneuroendocrinology* 23:485-495.

d. Durinx, C., Neels, H., Van der Auwera, J.C., Naelaerts, K., Scharpé, S., and De Meester, I. 2001. Reference values for plasma dipeptidyl-peptidase IV activity and their association with other laboratory parameters. *Clin. Chem. Lab. Med.* 39:155-159.

- These factors were considered in the design of the healthy donor trial and persons were randomly assigned to test and placebo controlled groups. We can provide additional details regarding the trial design in a revised manuscript. For the cHCV study, due to the challenges in recruiting sufficient patients, it was not possible to control for these factors.

4. Are the statements in the discussion justified regarding the significance of this study? There is probably some over-statement. This research group has shown that in mice DPP4 inhibition can increase the influx of lymphocytes into tumours. Is DPP4 inhibition likely to also increase systemic inflammation? , and thus exacerbate patients with inflammatory disorders. Several human cancers and rheumatoid arthritis have been associated with decreased rather than increased circulating DPP4 levels . The line of argument regarding the potential role of DPP4 inhibitors in autoimmune disease is unclear since long CXCL10, which is preserved by DPP4 inhibitors, is immunostimulatory.

- Based on the data presented in this manuscript and our group's recent work in mouse tumour models, we believe that our discussion related to the potential use of DPP4 inhibitors in cancer is justified. However, we accept the reviewer's point that the discussion regarding autoimmune disease is not justified in this context and therefore we will remove this text from the manuscript.

5. Overall, this study is incomplete. For instance, the detection range which is defined as between blank+3SD and blank+10SD only encompasses a single data point on the standard curves which goes against the principle of interpolation of sample data from standards data. Since the current study advocates for the potential application of sitagliptin in a therapeutic context, a rigorous examination of drug action in disease is warranted and unfortunately the disease group comprised only 3 patients which understandably did not permit any determination of statistical significance.

The authors propose that long CXCL10 increases at week 2 but the purported increase is detected in week 2 only and is not maintained into week 4 of the treatment window. Similarly, it is disappointing that the disease group was not monitored past week 2 and there is no data to support the reversibility of drug action upon withdrawal of therapy in this group.

- We apologize for the confusion here and removed the lines from the graphs in question. We used the measurement of blank+3SD and blank+10SD to define the limits of detection and limits of quantification of our assays, respectively. These are standard approaches for defining these parameters in immunoassays. We presented a single example but we have also tested the reproducibility of our assay. We believe that our study in healthy donors presents strong data for supporting further disease-based studies. As highlighted above, our aim was to recruit a larger cohort of cHCV patients but this was not possible.

Although the authors highlight that the levels of CXCL10 decrease during sitagliptin treatment, there also appears to be increased spread in the data during the treatment period [patient non-compliance?]. Inclusion of SD error bars are required for clarification.

- We preferred to include individual data points to reflect the real values obtained and present the actual distribution. The inclusion of SD error bars would require presentation of mean values instead of medians which does not seem the best choice for non-parametric data as noted in the comment below. We can however present IQR values to give a value to the differences in range and help clarification for the reader.

6. Statistics was appropriate. The use of Friedmans test was likely necessary because the data was non-parametric.

- We thank the reviewer for this comment.

2nd Editorial Decision

04 February 2016

Thank you for submitting your pre-revision PbP response to the referees for consideration in your appeal of our initial decision.

After careful examination of the arguments provided in your letter both by the editorial team and the referees, we feel that we can consider a major revision of your study, provided that you can make a strong case for the point 1 of referee 2 who seems to be particularly concerned about this. Seemingly, all other concerns raised by the referees must be convincingly addressed as you suggested in your letter.

Revised manuscripts should be submitted within three months of a request for revision; they will otherwise be treated as new submissions, except under exceptional circumstances in which a short extension is obtained from the editor.

Please note that it is EMBO Molecular Medicine policy to allow only a single round of revision and that, as acceptance or rejection of the manuscript will depend on another round of review, your responses should be as complete as possible.

Please read below for important editorial formatting.

I look forward to receiving your revised manuscript.

1st Revision - authors' response

19 February 2016

Referee #1 (Remarks):

Dipeptidyl peptidase 4 (DPP4) catalyzes the cleavage of two N-terminal amino acids from glucagon-like peptide 1 and several other proteins, including the chemokine CXCL10. Sitagliptin is a DPP4 inhibitor approved for treatment of type II diabetes. Many groups have reported that CXCL10 is increased in the blood of patients with chronic hepatitis C virus infection (cHCV), and the Albert group reported in 2011 that elevated CXCL10 was associated with the presence of a processed form similar to what would be expected after DPP4 cleavage. This shortened form of CXCL10 is believed to antagonize the chemotactic activity of the full-length form of CXCL10.

In this brief report, Decalf et al present the effects of sitagliptin treatment on CXCL10 levels and processing in 27 healthy subjects and 3 cHCV patients treated with sitagliptin, comparing them to 9 healthy subjects treated with a placebo. As predicted, sitagliptin treatment reduced DPP4-mediated processing of CXCL10 in both healthy and cHCV subjects. This result supports the claim that DPP4 activity contributes to CXCL10 cleavage in vivo. These preliminary human subject results are consistent with those recently published by the senior author in a mouse model (Nature Immunology 2015). The importance of the human result is that it provides a basis for human clinical trials in settings in which excess DPP4 activity and chemokine processing may impair immune function.

- We thank the reviewer for highlighting the importance of our results and their potential as a basis for future clinical trials.

Specific comments

1. Please indicate whether the cHCV patients included in this study were on other antiviral treatment.

- The patients included in our study were receiving pegylated Interferon + ribavirin, the standard of care for cHCV g4 at the time of recruitment, in addition to the experimental dose of sitagliptin. Additional details regarding treatment have been added to the methods and materials. As detailed in the cover letter, the approval of second-generation direct acting antivirals occurred much earlier than anticipated, with efficacy in chronic HCV g4 infected patients, thus obliging us to terminate our clinical trial prematurely. We have added text in the manuscript to make this point more clear.

2. I recognize that this report is quite preliminary. However, the inclusion of cHCV patients in this study begs the question of whether and how sitagliptin affects HCV viral load. This information would add to the novelty and interest of the report.

- The patients that were enrolled had failed prior type I IFN therapy and thus are considered hard-to-treat patients. In all three instances, patients failed to achieve an early virological response (EVR) as defined by a two-log reduction in virus at 12 weeks post-treatment. The

figure presented below shows the viral loads monitored over the course of our study. Although one patient showed a drop in viral load upon sitagliptin treatment, no major impact of sitagliptin on viral load was observed over this period. We have included these data as a new Expanded View Figure 2 in the revised manuscript.

3. In the abstract, lines 33-35, the authors write, "Sitagliptin treatment resulted in a significant decrease of CXCL103-77 concentration and a reciprocal increase in CXCL101-77, without affecting the total levels of the chemokine." This statement does not reflect the data in Figure 1C, showing a statistically significant reduction in total CXCL10 in healthy subjects during sitagliptin treatment. Indeed, in line 96 of the paper, the authors write, "Of note, levels of total CXCL10 were slightly decreased during sitagliptin treatment..."

- We thank the reviewer for highlighting this discrepancy and have now modified the manuscript accordingly. In order to assess the magnitude of the biological response to sitagliptin, we propose to include an effect-size analysis as detailed below:

Short CXCL10		Long CXCL10		Total CXCL10	
Groups	Cohen's d	Groups	Cohen's d	Groups	Cohen's d
Pre-Tx vs On-Tx	0.92	Pre-Tx vs On-Tx	0.27	Pre-Tx vs On-Tx	0.33
On-Tx vs Post-Tx	0.85	On-Tx vs Post-Tx	0.25	On-Tx vs Post-Tx	0.19
Pre-Tx vs Post-Tx	0.15	Pre-Tx vs Post-Tx	0.01	Pre-Tx vs Post-Tx	0.16

To perform this analysis, we compared the CXCL10 concentrations measured in all Pre-, On- and Post-sitagliptin samples. As shown, short CXCL10 was strongly affected by sitagliptin treatment and returned to baseline levels once therapy stopped. This phenotype was also true for long CXCL10, although to a lower extent. Size effect analysis showed a moderate influence of sitagliptin on total CXCL10 levels, supporting our interpretation that although long and short CXCL10 are affected by therapy, total levels of CXCL10 are mildly changed. We have included this analysis in the revised manuscript (and shown here for ease of review).

4. Figure 1C shows that sitagliptin has little or no effect on short CXCL10 levels in a number of healthy subjects. The authors may wish to comment on this. Did the same subjects show higher short CXCL10 levels at more than one time point?

- We thank the reviewer for highlighting this point. In order to provide the readership a critical point of view on our data, we have included a Expanded View Figure 1 to the revised manuscript (and presented below), which highlights the diversity of the healthy subjects' response to sitagliptin treatment. Based on the impact of sitagliptin on short CXCL10 levels, we defined three categories of subjects: responders (n=18), in which short CXCL10 levels are steadily undetectable upon sitagliptin treatment; partial responders (n=6), which show sporadic rise in short CXCL10 upon treatment; and non-responders (n=3), which show no change in short CXCL10 levels. We have also included in the figure the inhibition of DPP4 activity by sitagliptin in those subjects. We believe these data support the strong efficacy of sitagliptin on influencing circulating forms of CXCL10 but also emphasize the need to adjust therapy in subjects who fail to respond.

5. Were power calculations performed to support the usefulness of a study with only 3 cHCV patients?

- The original study was approved for an initial recruitment of 10 cHCV patients as the primary objective was safety assessment of the combined therapies. As detailed above the study was prematurely stopped due to ethical reasons related to approval of highly efficacious anti-viral therapies.

Referee #2 (Comments on Novelty/Model System):

Technical criticisms are below.

It is novel, but appears to be incomplete.

Medical impact is impaired by little disease data.

- We thank the reviewer for highlighting the novelty of our findings.

Referee #2 (Remarks):

CXCL10 (IP10) is an important substrate of the DPP4 enzyme and an important chemoattractant of T and NK cells. Activated T and NK cells have high levels of cell surface DPP4 (CD26).

Key authors of this ms derive from a research group that has shown in previous papers that DPP4 is an important enzyme in the degradation of CXCL10 in mice, and that DPP4 converts human CXCL10 into an antagonist of its cognate receptor CXCR3 in vitro. The enzyme activities of human and mouse DPP4 are believed to be the same. The biology of CXCL10 might differ between species.

The concentrations of DPP4 are elevated in human plasma in many chronic diseases, including HCV patients and some categories of cancer patients. Several human cancers have been associated with decreased rather than increased circulating DPP4 levels [review Gorrell 2001 Scand J Immunol]. This research group has shown in previous papers that the levels in human plasma of DPP4 and the DPP4-cleaved form of CXCL10 tend to correlate.

Comments:

1. The primary question that needs a rigorous investigation in such studies as this ms is where the processing of CXCL10 occurs . It could occur close to CXCR3-positive cells, in tissues, in circulating blood, or in blood or plasma after the blood is collected.

The most important, and readily accessible, ancillary question is whether the inhibition of CXCL10 degradation occurred entirely in vivo or to what extent the medicinally-derived DPP4 inhibitor that would have been in the blood at the time of blood collection caused the observed results.

To address this issue, Sitagliptin should have been added to all blood at the moment of blood collection in order to prevent any ex-vivo CXCL10 digestion. This is a confounder of data interpretation. DPP4 is very active at ambient temperature and has activity at 4 degrees, but not after freezing, so there is probably a correlation between the length of time after blood collection until the blood or plasma is frozen and the concentration of short-CXCL10.

An experiment should be performed to determine effects on CXCL10 of collecting blood into Sitagliptin. This data should be added to this ms.

- The reviewer is correct to highlight this important point. All blood samples in our studies were collected in P700 tubes, which contain DPP4 inhibitors. This information is described in the materials and methods, which we have now also highlighted in the results section of the revised manuscript. We also include for the reviewer a copy of the white paper from the manufacturer; detailing how P700 tubes preserve DPP4 substrates following blood draw (see attached). We validated these tubes in the lab and we are confident that our measurements reflect *in vivo* CXCL10 forms and not *ex vivo* (extracorporeal) cleavage events.

2. How many patients actually consumed the prescribed sitagliptin drug? Please present data on the concentrations of sitagliptin in the plasma samples. Such data will inform the reader on whether the concentrations of short/long CXCL10 that were measured correlate with the concentrations of sitagliptin in the blood samples.

- Data on DPP4 inhibition was previously presented (Price *et al*, CEI 2013) for the healthy donors and has now been included in the new Expanded View Figure 1 of the revised manuscript. As discussed above, of the 3 donors that consistently had higher levels of short CXCL10, one did not show DPP4 inhibition, perhaps reflecting human variability with respect to response to treatment. We have included additional information and text in the manuscript to highlight these outliers. For cHCV patients, we note the higher concentrations of DPP4 in the plasma. This, combined with the rapid *in vivo* clearance of sitagliptin made it difficult to accurately determine the impact on enzyme activity (Herman GA *et al*, *Clin. Pharmacol. Ther.* 78, 675–688, 2005).

3. *Early work on DPP4 measurements in human blood found that there are various causes of variability, including age, lower in women, increase in summer season, and decrease with depression which might or might not have been related to anti-depressant drugs. Were these parameters controlled in this study? Many HCV patients are depressed and prescribed anti-depressant drugs.*

References:

a. Maes, M., Scharpé, S., De Meester, I., Goossens, P., Wauters, A., Neels, H., Verkerk, R., De Meyer, F., P, D.H., Peeters, D., *et al*. 1994. Components of biological variation in prolyl endopeptidase and dipeptidyl-peptidase IV activity in plasma of healthy subjects. *Clin. Chem.* 40:1686-1691.

b. Maes, M., De Meester, I., Verkerk, R., Demedts, P., Wauters, A., Vanhoof, G., Vandoolaeghe, E., Neels, H., and Scharpé, S. 1997. Lower serum dipeptidyl peptidase IV activity in treatment resistant major depression - relationships with immune-inflammatory markers. *Psychoneuroendocrinology* 22:65-78.

c. Maes, M., Goossens, F., Lin, A., De Meester, I., Van Gastel, A., and Scharpé, S. 1998. Effects of psychological stress on serum prolyl endopeptidase and dipeptidyl peptidase IV activity in humans: higher serum prolyl endopeptidase activity is related to stress-induced anxiety. *Psychoneuroendocrinology* 23:485-495.

d. Durinx, C., Neels, H., Van der Auwera, J.C., Naelaerts, K., Scharpé, S., and De Meester, I. 2001. Reference values for plasma dipeptidyl-peptidase IV activity and their association with other laboratory parameters. *Clin. Chem. Lab. Med.* 39:155-159.

- These factors were considered in the design of the healthy donor trial and persons were randomly assigned to test and placebo controlled groups, and we have added text to the study design section to clarify these key points. For the cHCV study, due to the challenges in recruiting sufficient patients, it was not possible to control for these factors.

4. *Are the statements in the discussion justified regarding the significance of this study? There is probably some over-statement. This research group has shown that in mice DPP4 inhibition can increase the influx of lymphocytes into tumours. Is DPP4 inhibition likely to also increase systemic inflammation ? , and thus exacerbate patients with inflammatory disorders. Several human cancers and rheumatoid arthritis have been associated with decreased rather than increased circulating DPP4 levels . The line of argument regarding the potential role of DPP4 inhibitors in autoimmune disease is unclear since long CXCL10, which is preserved by DPP4 inhibitors, is immunostimulatory.*

- Based on the data presented in this manuscript and our group's recent work in mouse tumour models, we believe that our discussion related to the potential use of DPP4 inhibitors in cancer is justified. However, we accept the reviewer's point that the discussion

regarding autoimmune disease is not justified in this context and we have removed this text from the manuscript.

5. Overall, this study is incomplete. For instance, the detection range which is defined as between blank+3SD and blank+10SD only encompasses a single data point on the standard curves which goes against the principle of interpolation of sample data from standards data. Since the current study advocates for the potential application of sitagliptin in a therapeutic context, a rigorous examination of drug action in disease is warranted and unfortunately the disease group comprised only 3 patients which understandably did not permit any determination of statistical significance. The authors propose that long CXCL10 increases at week 2 but the purported increase is detected in week 2 only and is not maintained into week 4 of the treatment window. Similarly, it is disappointing that the disease group was not monitored past week 2 and there is no data to support the reversibility of drug action upon withdrawal of therapy in this group.

- We apologize for the confusion here and have removed the lines from the graphs in question. We used the measurement of blank+3SD and blank+10SD to define the limits of detection and limits of quantification of our assays, respectively. These are standard approaches for defining these parameters in immunoassays. We presented a single example but we have also tested the reproducibility of our assay. We believe that our study in healthy donors presents strong data for supporting further disease-based studies. As highlighted above, our aim was to recruit a larger cohort of cHCV patients but this was not possible.

Although the authors highlight that the levels of CXCL10 decrease during sitagliptin treatment, there also appears to be increased spread in the data during the treatment period [patient non-compliance?]. Inclusion of SD error bars are required for clarification.

- We preferred to include individual data points to reflect the real values obtained and present the actual distribution. The inclusion of SD error bars would require presentation of mean values instead of medians which does not seem the best choice for non-parametric data as noted in the comment below.

6. Statistics was appropriate. The use of Friedmans test was likely necessary because the data was non-parametric.

- We thank the reviewer for this comment.

Acceptance

16 March 2016

Please find enclosed the final reports on your manuscript. We are pleased to inform you that your manuscript is accepted for publication and is now being sent to our publisher to be included in the next available issue of EMBO Molecular Medicine.

***** Reviewer's comments *****

Referee #1 (Remarks):

The authors have addressed my concerns.

Referee #2 (Comments on Novelty/Model System):

This ms is greatly improved. Improvements made are likely all they can do.

Referee #2 (Remarks):

This ms is greatly improved. Improvements that were not colored red were noted, as well as those that were colored red.

It now provides a novel document that will inform the readers .

However, I believe that randomising patients is insufficient to control for the many sources of variability in a small cohort. Multivariate analysis should also be performed.

Corresponding Author Name: Darragh Duffy
Journal Submitted to: EMBO Molecular Medicine
Manuscript Number: EMM-2015-06145